# Development of a Framework for Scaling Up Community-Based Health Promotion: A Best Fit Framework Synthesis

**DOI:** 10.3390/ijerph19084773

**Published:** 2022-04-14

**Authors:** Philipp Weber, Leonie Birkholz, Simone Kohler, Natalie Helsper, Lea Dippon, Alfred Ruetten, Klaus Pfeifer, Jana Semrau

**Affiliations:** Department of Sport Science and Sport, Friedrich-Alexander Universität Erlangen-Nürnberg, Gebbertstr. 123b, D-91058 Erlangen, Germany; leonie.birkholz@fau.de (L.B.); simone.kohler@fau.de (S.K.); natalie.helsper@fau.de (N.H.); lea.j.dippon@fau.de (L.D.); alfred.ruetten@fau.de (A.R.); klaus.pfeifer@fau.de (K.P.)

**Keywords:** health promotion, health equity, scaling up, community, physical activity promotion

## Abstract

Community-based health promotion with a focus on people with social disadvantages is essential to address persistently existing health inequities. However, achieving an impact on public health requires scaling up such approaches beyond manifold funded pilot projects. The aim of this qualitative review is to provide an overview of scaling-up frameworks in health promotion and to identify key components for scaling up community-based health promotion. First, we conducted a systematic search for scaling-up frameworks for health promotion in PubMed, CINAHL, Scopus, Web of Science, PsycInfo, and SportDiscus. Based on the included frameworks, we created an a priori framework. Second, we searched for primary research studies in the same databases that reported scaling-up processes of community-based health promotion. We coded the data using the a priori framework. From 80 articles, a total of 12 frameworks were eligible, and 5 were included for data extraction. The analysis yielded 10 a priori defined key components: “innovation characteristics”; “clarify and coordinate roles and responsibilities”; “build up skills, knowledge, and capacity”; “mobilize and sustain resources”; “initiate and maintain regular communication”; “plan, conduct, and apply assessment, monitoring, and evaluation”; “develop political commitment and advocacy”; “build and foster collaboration”; “encourage participation and ownership”; and “plan and follow strategic approaches”. We further identified 113 primary research studies; 10 were eligible. No new key components were found, but all a priori defined key components were supported by the studies. Ten key components for scaling up community-based health promotion represent the final framework. We further identified “encourage participation and ownership” as a crucial component regarding health equity.

## 1. Introduction

The community, defined as a geographic area [1], is a central setting for health promotion [2]. Community-based health promotion interventions [3,4] could offer a number of benefits, such as influencing contextual factors [2] or “creating healthy community environments through broad systemic changes in public policy and community-wide institutions and services” [5] (p. 530). Moreover, interventions focusing on the community are expected to make substantial contributions to health equity, as the community affects the entire population. Health equity means that “everyone should have a fair opportunity to attain their full health potential and that no one should be disadvantaged from achieving this potential” [6] (p. 4).

However, as community-based health promotion interventions generate a population-wide impact and are intended to address all people equally, there is a risk of reaching particularly socially advantaged population groups. This can lead to more health inequities, referring to the inequality paradox [7]. To address this issue, community-based interventions should be implemented and sustained with a focus on deprived communities and people with social disadvantages (e.g., very low income, very low school education, or single parents). This has been shown in several studies, which have indicated that community-based health promotion interventions with this focus can be effective in addressing health inequities [3,8].

A prerequisite to achieving a recognizable public health impact, along with these community-based health promotion interventions, is to enable scaling up beyond extensively funded model projects. However, few interventions have been scaled up thus far. For instance, only 40 public health interventions currently show evidence of being scaled up [9,10]. We define scaling up according to the WHO’s definition as “deliberate efforts to increase the impact of successfully tested health innovations so as to benefit more people and to foster policy and program development on a lasting basis” [11] (p. 2). “Health innovation” refers to the intervention itself, which is new or perceived as new, and the actions required for its successful implementation, resulting in a set of interventions [11].

Scaling up also raises the challenge of taking health equity into account [12]. Therefore, in our understanding, scaling up is not just about reaching more people through implementation on a larger scale; it is also about creating healthy environments and fostering sustainable implementation that is capable of addressing this challenge. This perspective is also reflected in the definition of scaling up: “foster policy and programme development on a lasting basis” [11] (p. 2), which points to the importance of institutional capacity-building and sustainability.

For example, the research project VERBUND (Scaling up and cooperative implementation of community-based physical activity promotion) aims to develop and test a concept to scale up a complex community-based physical activity promotion intervention [13,14] and to sustainably embed community-based physical activity promotion in Germany. Following the VERBUND project, this article therefore considers physical activity as an area of interest for health promotion, emphasizing the close connection between physical activity and the concept of health promotion [15].

A growing number of frameworks for scaling up health innovations in the field of health promotion have been published [10,11,16,17,18,19,20,21,22], although they are mostly unrelated to health equity and focus on implementation science. Greenhalgh and Papoutsi [23] described three different strategies through which spread and scale-up can be approached: mechanistically (implementation science), ecologically (complexity science), and socially (social science). Complexity science and implementation science can provide different perspectives for understanding the mechanisms underpinning the scaling-up process [23]. However, we emphasize the importance of a social science perspective [24] for scaling up because it could “help researchers […] to tap into (with a view to influencing) the organisational and societal influences” [23] (p. 4) regarding scale-up and health equity.

In summary, scaling up health innovations beyond model projects is crucial for public health. Despite the growing number of scaling-up frameworks, to our knowledge, existing frameworks or models consider health equity issues insufficiently in the process of scaling up community-based health promotion. Therefore, the first aim of this qualitative review is to compile a broader overview of scaling-up frameworks for health innovations in health promotion. The second aim is to identify key components for scaling up within these frameworks and validate the results with the findings of primary research studies that describe the scaling-up processes of health innovations in the field of community-based health promotion.

## 2. Materials and Methods

The best fit framework synthesis approach was chosen based on the exploratory nature of this review [25]. This approach enables the authors to systematically identify relevant frameworks, models, or theories for scaling up health innovations and create an a priori framework with the “best fit” to the topic. Further evidence derived from primary research studies is coded against this a priori framework [26]. Thus, theoretical and science-based perspectives are combined into key components for scaling-up processes.

### 2.1. Search Method Frameworks

A systematic search strategy was used to identify frameworks, models, concepts, and theories that report on scaling-up processes. This search was based on the behavior of interest, health context, exclusion, and models or theories (BeHEMoTh) approach [27] as suggested by Carroll et al. [25]. The BeHEMoTh strategy was modified to define public health, health innovations, or health promotion, including physical activity, as the behavior of interest. The health context included the phenomenon of scaling up. The final search strategy can be found in the Appendix A. Using this approach, the PubMed, CINAHL, Scopus, Web of Science, PsycInfo, and SportDiscus databases were screened systematically in January 2021 for eligible publications by two researchers (P.W. and B.G.). No restrictions on the publication date were set. Moreover, gray literature was searched with related search terms using Google Scholar. Finally, literature related to scaling up and health promotion including physical activity promotion was included. Articles referring to infectious diseases, primary health care, and technical innovations, or those limited to a special population were excluded (see Appendix A). Two reviewers (P.W. and B.G.) independently screened the titles, abstracts, and full texts for congruence with these inclusion criteria using EndNote software, Philadelphia, PA, USA. Unclear publications were discussed by three authors (P.W., B.G., and J.S.), and consensus regarding inclusion or exclusion was reached.

### 2.2. Creation of the a Priori Framework

The identified scaling-up frameworks were used as the basis for the development of an a priori framework with the best fit for our topic using thematic analysis techniques [27]. In the first step, two authors (P.W. and J.S.) reviewed the frameworks separately to familiarize themselves with the data. In the second step, the two researchers inductively generated codes using MAXQDA software, Berlin, Germany, based on the similarities and differences between the frameworks [25]. The identified codes were then collated into potential themes. Both reviewers again reviewed and discussed the content of the potential themes and the data sets and refined them, if necessary. In this way, consensus was reached on a set of defined themes representing the core ideas of the scaling-up frameworks. Labels and short definitions were added to the themes based on the codes. The a priori framework used for further analysis was created using these themes.

### 2.3. Search Strategy for Primary Research Studies

To identify relevant primary research studies that report scaling-up processes in the field of community-based health promotion, including physical activity promotion, a second systematic literature search was conducted. As the aim was to identify key components for scaling up and present them using scientific evidence from the studies, we applied a broad search focus for primary research studies. PubMed, CINAHL, Scopus, Web of Science, PsycInfo, and SportDiscus were systematically searched by two authors (P.W. and L.B.), and a gray literature search was conducted using Google Scholar. The search terms were developed in cooperation with a library search specialist and contained keywords for scaling up, community, and health promotion (see Appendix A). The search was conducted in February 2020 with no restrictions on publication dates. The studies were screened using EndNote software, Philadelphia, PA, USA, and duplicates were removed. Titles and abstracts were screened based on the inclusion and exclusion criteria. The inclusion criteria comprised health promotion or physical activity promotion interventions and a report of the scaling-up process. Articles focusing on primary health care, technical innovations, or infectious diseases were excluded, as well as those centered solely on personal health-related outcomes (see Appendix A). Two reviewers (P.W. and L.B.) independently screened the full texts and discussed the results with the author team to reach a consensus on the final inclusion of primary research studies.

### 2.4. Data Analysis and Synthesis

For the data analysis and synthesis, we followed the approach described by Carroll et al. [28]. After the studies were selected, the data were managed using MAXQDA software, Berlin, Germany. Therefore, a data extraction form was developed that included the themes of the a priori framework [26]. The data in the studies were extracted from the Results and Discussion sections and consisted of verbatim quotations from the study participants or findings and interpretations by the authors that were clearly supported by the data [25]. The quotes referred to factors that influenced the scaling-up process, either positively or negatively. Two researchers independently coded two studies to agree on the final form. After clarification and discussion, the researchers each extracted data from approximately half of the studies that had not yet been coded. In this way, the results were coded using the data extraction sheet following a deductive approach. If the codes did not fit the data extraction form, they and the themes were then discussed by the reviewers, and when necessary, new themes were created that fit the codes uncovered by the a priori framework. This process resulted in a final list of themes agreed upon by the author team. In the final step, the themes were synthesized with reference to the data extracted from the studies and the frameworks; furthermore, the existing labels and short definitions were refined. In addition, the author team discussed the potential relationships between the themes and the possible factors behind these themes.

### 2.5. Quality Appraisal and Sensitivity Analysis

During the data extraction, two reviewers completed a quality assessment of the studies [28] using a pragmatic quality assessment checklist [29]. This assessment contained four questions that addressed relevant factors of the research to evaluate whether the publications clearly described the research question and study design, the method for recruiting or selecting participants, and the data collection and analysis methods [29]. In the assessment, the researchers first independently scanned two studies and extracted appropriate text passages that were generally found in Introduction and Methods sections. The question responses were “yes”, “no”, or “unsure”. After reaching a common understanding and comparing the results, each researcher analyzed half of the remaining studies. The researchers then examined the extracted text passages and the original text of each other’s studies, and any disagreements that arose were discussed and resolved by consensus. Studies were then classified as adequately reported if they provided sufficient information about two or more criteria. Inadequately reported studies were not excluded from the analyses.

Therefore, a sensitivity analysis was conducted to explore whether the inadequately reported studies or other characteristics had a substantial effect on the results [29]. To check whether this broad focus in the search for primary studies had an influence, characteristics such as country, setting, intervention type, or whether the studies were based on scaling-up frameworks were also assessed. The final themes derived from the primary research studies were reviewed to identify any potential influences of these factors. 

## 3. Results

### 3.1. Study Selection

The BeHEMoTh search approach focusing on scaling-up frameworks and guidelines resulted in 7765 citations retrieved from all databases searched and 8 articles derived from a manual search of gray literature. After duplicates were removed, 4657 unique citations were identified, of which 4557 were excluded with title and abstract screening. The full text of 80 articles was screened, and 68 were excluded because they did not contain a framework (*n* = 34) or for other reasons (see Figure 1). Twelve eligible frameworks were identified, but we focused our analysis on frameworks with rich information and an additional relation to health equity and/or physical activity, as these reflected major aspects of our purpose. Therefore, five scaling-up frameworks were included in the analysis. The details of the frameworks can be found in Table 1.

The systematic literature search for primary research studies led to 9352 citations and 9 additional references identified by the gray literature search. Of 6297 unique studies, 6189 were excluded with title and abstract screening. For 113 articles, full texts were retrieved; 103 were excluded because they were not related to scaling up (*n* = 33), not located in the field of health promotion (*n* = 33), not primary research studies (*n* = 26), or focused on health-related results or located in primary health care (*n* = 2) (see Figure 2). The characteristics of the included studies are illustrated in Table 2. The assessment of reporting quality of the primary research studies indicated that all citations were adequately reported, as they were rated with two or more agreed quality statements (see Appendix A). Half of the included studies met all quality appraisal criteria; only one met two of the four criteria.

### 3.2. The Framework

The framework, including 10 key components, was created by reviewing 5 scaling-up frameworks and 10 primary research studies. As described in the Section 3, 10 themes were generated and finalized through the analysis of the frameworks using an inductive approach, representing an a priori framework. Subsequent deductive coding of the primary research studies using these a priori themes revealed that all components were supported by these studies, and no further themes were identified inductively. Instead, more in-depth and supplementary information on the themes was obtained. The final themes represent key components that are crucial for scaling up health innovations in the field of health promotion, representing our framework. As no chronological order was identified, and due to dynamic interactions and mutual dependencies between the components, they are displayed in a circular array, which can be seen in Figure 3. The following descriptions of the key components are based on representative examples from the articles and contain the core ideas and perspectives.

#### 3.2.1. Innovation Characteristics

The innovation characteristics embody the center of the scaling-up processes, as every action realized for scaling up is connected to the innovation. For instance, the WHO pointed out that “resisting pressure to rapidly scale up the innovation before its effectiveness and feasibility have been fully established is essential” [11] (p. 19). Beyond effectiveness, the innovation should be based on substantial theory and evidence [38]. This further strengthens users’ beliefs in the innovation’s benefits, which is, moreover, cited as an important characteristic [39]. In addition, the problem addressed by the innovation should be perceived as relevant by its users [41]. Additionally, if the design is simple, and there are “no dissenting views, scale-up is much more likely to happen” [22] (p. 2). In line with this, the innovation needs to be flexible in the context, so single components of the innovation’s “set of interventions” can change and diverge over time [11].

#### 3.2.2. Clarify and Coordinate Roles and Responsibilities

During the scaling-up process, different actors (e.g., individuals or organizations) will be involved. A major task in scaling up health innovations is to define relevant roles and their responsibilities ab initio and to keep them transparent throughout the whole pathway [32]. We identified different roles that appear to be important in scaling-up processes, such as local implementers, champions, and implementation supporters. Some of these roles can be formal (e.g., the local implementer as a formal project coordinator) or informal (e.g., champions). Champions appear to be valuable in scaling up health innovations, as they are “usually the gatekeepers to their entire organization’s commitment” [40] (p. 283). In our findings, researchers often acted as implementation supporters, building up skills and knowledge [35,38,39].

#### 3.2.3. Build Up Skills, Knowledge, and Capacity

Personal, technical, and organization-related skills and knowledge are crucial factors to successfully scale up that were supported by all primary research studies. This is important for all actors involved in the process, not only those engaged in the local implementation. Networking with target groups, stakeholders, or policymakers and leadership, for instance, was cited, among others, as being a relevant skill, as well as “managerial expertise and skills in advocacy” [11] (p. 18). Skills and innovation-specific knowledge were promoted, for example, through training sessions or workshops and through supporting materials. Some authors have pointed out that the possibility of consultation at a competence center was found to be very helpful [35]. To ensure the benefit of the scaling-up process, a high-quality standard for workshops, training sessions, and supportive materials should be established [34].

#### 3.2.4. Mobilize and Sustain Resources

The availability of financial and personal resources was a common component discovered in most primary research studies. To start the scaling-up process, an adequate amount of seed funding is valuable for attracting new implementation sites for the innovation, as it “facilitated the decision-making of the local administrators to implement the project” [35] (p. 4). To sustain funding for the innovation, it must be linked to macro-level funding mechanisms, and it is critical for identifying existing “sources of support at the national or district level” [11] (p. 28). Another option for achieving funding sustainability is to ensure diversification through different funding providers. In addition to funding challenges, the availability of a skilled workforce has been reported as a barrier [34]. One reported challenge was finding the right personnel and dealing with staff turnover [41].

#### 3.2.5. Initiate and Maintain Regular Communication

To build relationships and disseminate information, communication structures and channels must be established. For successful communication, the structures should be adapted close to the target groups and thus strengthen the relationships [11]. Therefore, different communication channels, especially electronic ones such as newsletters, news media, and websites, can be used for this purpose [40]. Some researchers have pointed out that in addition to electronic communication, face-to-face communication at meetings, conferences, or workshops is essential for building formal and informal relationships [11].

#### 3.2.6. Plan, Conduct, and Apply Assessment, Monitoring, and Evaluation

The literature indicated that collecting relevant local data, specifically at the beginning, is important for understanding the local perceived needs, procedures, and administrative enablers and barriers to successful scaling up [33]. For this purpose, appropriate assessment tools are crucial [37]. One study specifically recommended the use of an advisory board [42]. However, it is important to perform an assessment not only at the beginning but also to conduct ongoing monitoring and evaluation, which are necessary for carrying out the needed adjustments in time to optimize the innovation [41]. This was also highlighted in a study by Nigg et al., who described the “awareness of program weaknesses and needed adjustments were achieved through ongoing evaluations, creating a constructive feedback loop for continuous improvement” [40] (p. 281).

#### 3.2.7. Evolve Political Commitment and Advocacy

Several frameworks and scholars have implied that political commitment and advocacy play an important role in scaling up health innovations. This support includes policy commitment and advocacy at the national, regional, and local levels as well as the commitment of individuals or organizations, such as governments or nongovernment organizations (NGOs) [22]. Thus, a policy or guideline should be in place to legitimize the innovation. This is a success factor that leads to a dedicated budget, resources, goals, and the capacity to translate policy into action [34]. Nonetheless, one study indicated that there is a difference between having political support from politicians and having established policies in place, which is well demonstrated in the following example: “at the local level, Recreovía and HEVS were legitimized by their inclusion in local development plans or administrative acts. This helped protect them from changes due to administration cycles” [34] (p. 48). This example also illustrates that political commitment and advocacy should be constantly renewed and ensured, especially with regard to sustainability.

#### 3.2.8. Build and Foster Collaboration

The importance of strengthening existing partnerships and building new ones was emphasized by several studies as being relevant for scaling up [31,42]. Accordingly, the involvement of local partners should be considered at an early stage in order to gain important insights and develop a common approach and understanding [31,37]. This phenomenon was described by Nigg et al. [40] (pp. 283–284):

One of the main reasons for our successful recruitment was to include the three major providers from the start-during the conceptualization discussions and in the pilot. Thus, the organizations knew what would be involved. Those early successes showed the value added by implementing Fun 5, motivated a private provider and a couple of DOE (Department of Education) districts in the first year to implement across their sites.

However, this includes public and non-profit organizations from different sectors [32] as well as cooperation with the research community [33]. The resulting collaborations are useful and necessary to reduce costs, ensure ongoing support, expand services, and support logistics [34,37].

#### 3.2.9. Encourage Participation and Ownership

Active participation, co-creation, and co-ownership are relevant factors in scaling-up efforts [35,37,42]. Co-creation can be described as a process of collaborative development that connects key stakeholders to collaboratively achieve a result [37]. Studies have reported that this has played an important role, particularly in projects of longer duration [37]. It should be considered throughout the whole process that “the active participation of the community in planning, implementing, and monitoring interventions is widely cited as a crucial factor” [22] (p. 3). This means involving all relevant actors, such as community members, community groups, local experts (e.g., sport club representatives), or local government departments [33,35,37,38]. Participation pools resources, expands capacity, and strengthens partnerships [11]. Moreover, participation provides acceptance and relevance for the initiative [37].

#### 3.2.10. Plan and Follow Strategic Approaches

An intervention that is intended to be scaled up cannot be implemented without planning. Therefore, a phased approach should first be used to test the intervention in a pilot setting [11,22]. Then, the intervention can be expanded, adjusting to the context each time, which was recommended by Yamey [22] (p. 2): “A related concept is the notion of going to scale in a phased manner beginning with a pilot program, followed by stepwise expansion, learning lessons along the way to help refine further expansion.” For this approach, the required time needs to be considered [11]. In addition, a common understanding among all partners involved is beneficial for strategic alignment [37]. Integrating the innovation into existing structures and systems (e.g., health care systems) should be considered, and as relevant milestones are reached, they should be noticed and celebrated [37].

### 3.3. Sensitivity and Thickness

A sensitivity analysis was conducted to assess whether study characteristics such as reporting quality, country, setting, intervention type, or the theoretical ground of the primary research studies had a potential influence on the results (see Appendix A). For all key components, thick data were found. All key components were covered by data from at least six primary research studies. No study was based on a scaling-up framework or model included in the synthesis for the a priori framework. Moreover, none of the study characteristics described above biased the creation of the final 10 key components because no key component was supported exclusively by studies that shared one of these characteristics. No formal procedures for identifying disconfirming cases were performed because numerous cases of dissonance, such as contradictory views, were recognized within most key components [25]. For example, adequate funding was reported to be a facilitator for scaling up, whereas a lack of funding was commonly described as a barrier.

## 4. Discussion

The aim of this qualitative review was to provide an overview of scaling-up frameworks for health innovations in health promotion and to identify key components for scaling up health innovations in the field of community-based health promotion. For this purpose, we used the best fit framework synthesis and systematically identified 12 frameworks in the field of health promotion. We analyzed five of these frameworks that considered health equity or physical activity in terms of key components to create an a priori framework. This analysis yieded 10 a priori defined key components: innovation characteristics; clarify and coordinate roles and responsibilities; build up skills, knowledge, and capacity; mobilize and sustain resources; initiate and maintain regular communication; plan, conduct, and apply assessment, monitoring, and evaluation; evolve political commitment and advocacy; build and foster collaboration; encourage participation and ownership; and plan and follow strategic approaches. These key components depict the proposed framework and are further supported by 10 primary research studies describing scaling-up processes in the field of community-based health promotion. As these key components demonstrate the complexity that prevails in scaling-up processes, mutual interactions and dependencies between them must be noted. These interactions could be seen, for instance, in the category “build and foster collaboration” that assumes to “initiate and maintain regular communication” to get in contact with relevant stakeholders for initiating and sustaining collaborations. Another example is the category “plan and follow strategic approaches”, which is important for the entire scaling-up process. The scaling-up approach itself must be strategically planned and implemented, but every decision made pertaining to other key components should also be strategically considered. Finally, the key components will be incorporated to develop a science- and practice-based concept for scaling up a community-based physical activity promotion intervention, together with insights from stakeholders at the national and community levels in Germany, in the context of the research project VERBUND.

In summary, most of the 10 key components were frequently mentioned in the analyzed frameworks. For instance, the key component “innovation characteristics” is widely cited in scaling-up frameworks. It appears as part of the “CORRECT” attribute list in the WHO ExpandNet [11] and was proven to be part of an essential scaling-up pathway [43]. Regarding the components “plan and follow strategic approaches” and “plan, conduct, and apply assessment, monitoring, and evaluation”, specific aspects of assessment and evaluation appear frequently [16,20,43]. Additionally, political support and financial resources were important and seem to be commonly accepted and relevant components, which were often mentioned by most of the identified frameworks [17,43]. Moreover, Koorts et al. [43] cited political support from policymakers as a crucial factor in scaling-up processes, which is thus represented in the key component “evolve political commitment and advocacy”.

As stated in the introduction, the aim of the research project VERBUND is to develop and test a concept to scale up a physical activity promotion intervention and sustainably embed community-based physical activity promotion in Germany. Therefore, we focused on a social science perspective on scaling up and referred to the interplay of structure and agency in health promotion [24]. This provides a contrast with other frameworks that tend to be concerned with aspects of dissemination or large-scale implementation of a health promotion intervention, which is crucial in our understanding but only one part of scaling up. This is also reflected within the perspectives of implementation science, complexity science, and social science, as described by Greenhalgh and Papoutsi [23]. From the implementation science perspective, scaling up refers to a “structured and phased approach to developing, replicating, and evaluating an intervention in multiple sites” [23] (p. 1). Complexity science shifts the focus to the system in which scaling up occurs, highlighting the self-organizing and dynamic characteristics and emphasizing the need for a flexible and adaptive approach [44,45]. Our understanding of scaling up could explain why we identified the component “participation and ownership”, which is more implicitly stated in some other frameworks. The WHO ExpandNet [11] and the framework published by Yamey [22] consider participation valuable—for example, for planning, implementing, and monitoring interventions or for mobilizing support and addressing local needs—whereas in other frameworks [10,31,32] participation is mentioned less. In the context of tackling health inequities, participatory approaches and the involvement of different actors throughout the whole process have been proven to be crucial [35,46,47,48]. Particularly regarding policymakers and professionals, participatory approaches can promote systematic, health-promoting policy changes [49].

As the best fit framework synthesis approach aims to create a new framework or conceptual model for a special purpose [25], the question arises as to whether the key components can be seen as a new framework. Scaling-up frameworks or models should offer a structure for organizing scaling-up processes. Moreover, frameworks or models should be able to inform decisions and judgments arising during scaling-up processes [50]. Regarding the resulting key components, especially compared with other frameworks [11,31,32], they do not provide guidance for how to proceed in a timely manner for scaling up. Even though some key components should clearly appear prior to others, we cannot show a chronological order in our framework. Through the inductive approach, a temporal component could only be identified for some key components, which is incorporated within them. The key component “Plan, conduct, and apply assessment, monitoring, and evaluation” emphasizes the importance of timing—e.g., starting with an assessment at the beginning of the process. For some other key components, no clear temporal alignment across frameworks could be identified. Although an intuitive chronological order seems possible, it cannot be clearly based on the presented data and may be dependent on many other contextual factors. Furthermore, since we were also unable to identify any direct relationships between the components, it seems obvious that these findings demonstrate no model or implementation guidelines. Nevertheless, these key components can be used as a basis for decisions during scaling-up processes. Therefore, we consider the key components as a framework [51,52] that needs to be further elaborated with practice-based evidence. Future research could moreover aim to embed the key components in a timeframe.

The best fit framework synthesis approach was chosen to conflate the insights of scaling-up frameworks with evidence derived from primary research studies on this topic. The approach builds on existing frameworks, but with a special aim at a new, relevant population [25]. We modified this approach slightly, as we did not focus on a new population but on a specific perspective for scaling up. However, we were able to apply the described best fit framework approach [25] to our objectives. Nevertheless, the procedure revealed some limitations of the synthesis approach. In our systematic searches, we focused on scaling health promotion innovations, as this is the focus of this review. Thus, the inclusion criteria were wide-ranging, leading to many possible eligible citations. We decided to focus on articles with rich information and related to either health equity or physical activity. Therefore, 5 of 12 articles reporting frameworks were used for data extraction. In addition, because of the expected small number of primary research studies on scaling up health promotion interventions [9,10], we decided not to use health equity as a search term or inclusion criterion. However, it cannot be ruled out that a closer focus on health equity through strict inclusion criteria for interventions that have been proven to contribute to health equity would lead to different findings. As a deeper understanding is needed for the sustainable scaling up of community-based health promotion with focus on health equity, this framework can be seen as a step towards this direction. As the a priori defined themes were derived from frameworks located in the field of health promotion, this may have contributed to the fact that no differences in the sensitivity analysis were found regarding the key components. However, this may not imply that scaling-up processes are independent of factors such as setting or intervention characteristics, as studies with different preconditions show different manifestations of the key components. For example, the key component “build and foster collaboration” requires different actors or organizations depending on the intervention characteristics and the local preconditions.

In addition to these differing aspects of the primary research studies, we included only scientific literature. All of the studies were based on scientific cooperation during the scale-up, whereas there might be successfully scaled-up health promotion innovations that are not. This practice-based evidence could provide more in-depth insights into scaling-up processes. Accordingly, in the research project VERBUND, the science-based insights will be amplified through practice-based evidence derived from the workshops with the national and community shareholders.

Regarding the methodology of the best fit synthesis approach, we followed the instructions as closely as possible and described our procedures in detail above, although it was challenging, as the available literature does not provide detailed instructions for the coding process within the best fit framework synthesis. A more in-depth thematic analysis approach, such as reflexive thematic analysis [53], could be valuable for this approach and lead to a more in-depth understanding of the scaling-up process. However, despite the merit of reflexive thematic analysis, its use within the elaborate best fit framework synthesis approach must be balanced against the available time and personnel resources.

## 5. Conclusions

We identified 12 scaling-up frameworks in the field of health promotion. Based on 5 frameworks that considered health equity or physical activity promotion, we further identified 10 key components that represent our final framework for scaling up community-based health promotion. This framework is supported by the findings of 10 primary research studies. In the analysis of the frameworks and studies, we further identified “encourage participation and ownership” as an important key component that contributes to the development of partnerships and the acceptance of initiatives for scaling up community-based health promotion with a focus on health equity. Our framework offers a helpful orientation for planning and implementing scaling-up approaches in the field of community-based health promotion. However, we were not able to derive a concrete step-by-step procedure. This requires the development of a specific scaling-up concept that considers practice-based evidence based on the experience and knowledge of various national and community actors.

## Figures and Tables

**Figure 1 ijerph-19-04773-f001:**
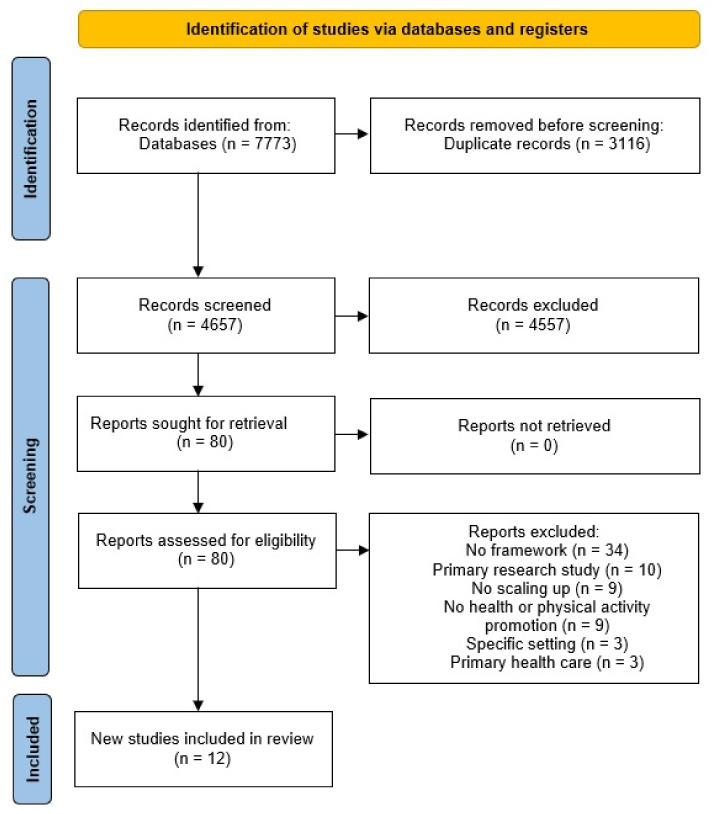
PRISMA flow diagram of the BeHEMoTh search for frameworks.

**Figure 2 ijerph-19-04773-f002:**
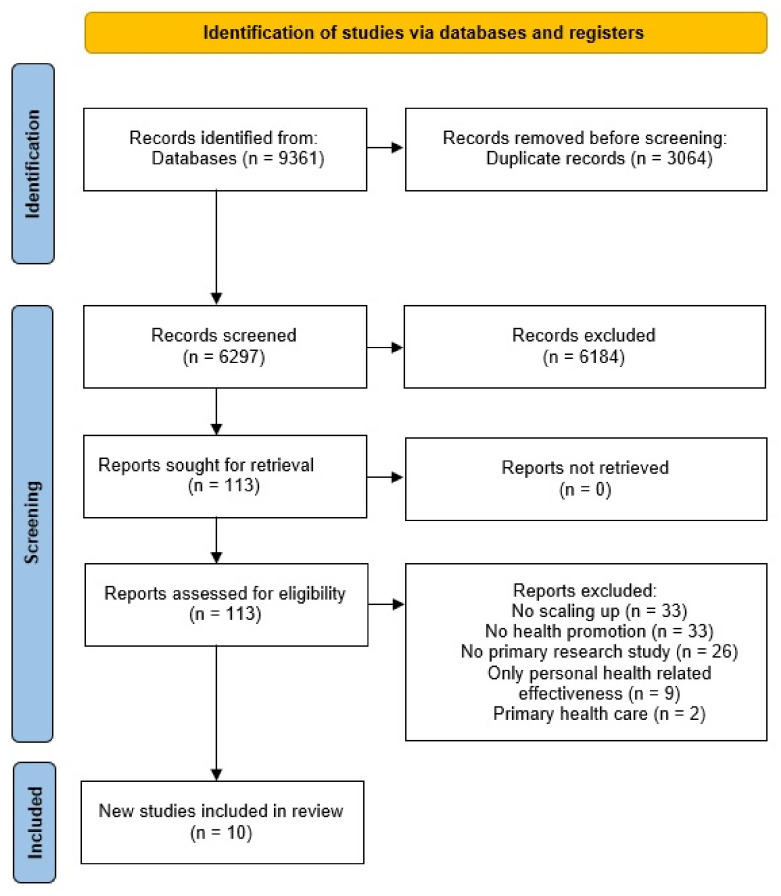
PRISMA flow diagram of the search for primary research studies.

**Figure 3 ijerph-19-04773-f003:**
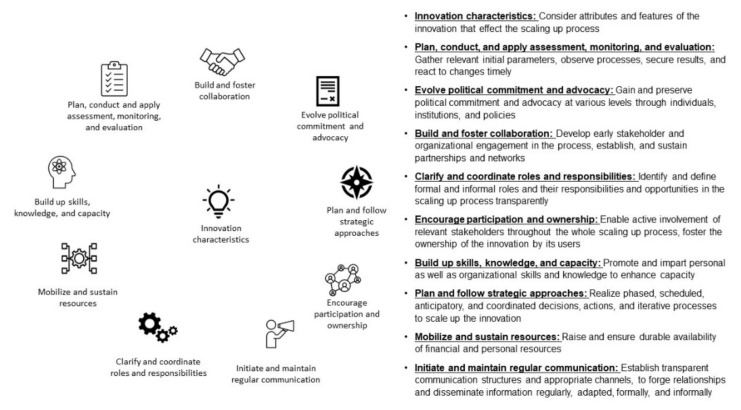
Scaling-up framework with key components for scaling up community-based health promotion.

**Table 1 ijerph-19-04773-t001:** Characteristics of the identified frameworks.

Authors	Year	Title	Journal	Included in Analysis	Details
Bauman A.E., Nelson D.E., Pratt M., Matsudo V., Schoeppe S. [18]	2006	Dissemination of physical activity evidence, programs, policies, and surveillance in the international public health arena	American Journal of Preventive Medicine	No	A six-step framework for understanding the attributes of successful international dissemination.
Cooley L., Kohl R. [30] -identified in gray literature search-	2006	Scaling up—from vision to large-scale change: a management framework for practitioners	Washington, DC: Management Systems International	No	The Scaling Up Management Framework provides 3 steps with 10 tasks for scaling up from the discipline of “strategic management”.
Baker E.L. [16]	2010	Taking programs to scale: a phased approach to expanding proven interventions	Journal of Public Health Management and Practice	No	A 5-phase approach that provides tasks—e.g., for clarity of roles and responsibilities during the process.
World Health Organization [11] -identified in gray literature search-	2010	Nine steps for developing a scaling-up strategy	WHO Press, World Health Organization	Yes	A 9-step guide intended for program managers and others planning to scale up successfully tested interventions.
Yamey G. [22]	2011	Scaling up global health interventions: a proposed framework for success	PLoS Medicine	Yes	Success factors for scaling up were grouped into six categories that apply to represent different components of the scaling up process.
Milat A.J., King L., Bauman A.E., Redman S. [21]	2013	The concept of scalability: increasing the scale and potential adoption of health promotion interventions into policy and practice	Health Promotion International	No	Intervention and research design factors that have the potential to “scale up” interventions.
Barker P.M., Reid A., Schall M.W. [17]	2016	A framework for scaling up health interventions: lessons from large-scale improvement initiatives in Africa	Implementation Science: IS	No	The framework describes a sequence of activities required to fully implement a program at scale, the mechanisms that facilitate adoption, and the underlying factors and support systems.
Milat A.J., Newson R., King L., Rissel C., Wolfenden L., Bauman A., et al. [20]	2016	A guide to scaling up population health interventions	Public Health Research & Practice	No	The practical framework includes 4 steps to assist, for example, health policy makers in scaling up effective population health interventions and to help design research studies.
Reis R.S., Salvo D., Ogilvie D., Lambert E.V., Goenka S., Brownson R.C., Lancet Physical Activity Series 2 Executive Committee [10]	2016	Scaling up physical activity interventions worldwide: stepping up to larger and smarter approaches to get people moving	The Lancet	Yes	Scaling-up specific adaptation of the RE-AIM framework to improve efforts to develop, implement, and evaluate physical activity interventions.
Koorts H., Eakin E., Estabrooks P., Timperio A., Salmon J., Bauman A. [31].	2018	Implementation and scale up of population physical activity interventions for clinical and community settings: the PRACTIS guide	The International Journal of Behavioral Nutrition and Physical Activity	Yes	This framework provides 4 iterative steps for effectively planning the implementation and scaling up of physical activity interventions.
Fagan A.A., Bumbarger B.K., Barth R.P., Bradshaw C.P., Cooper B.R., Supplee L.H., Walker D.K. [19].	2019	Scaling up evidence-based interventions in US public systems to prevent behavioral health problems: Challenges and opportunities	Prevention Science	No	This framework identified a set of factors that affect scaling up in 5 public systems and provides actions needed to significantly increase scaling up.
Nguyen D.T.K., McLaren L., Oelke N.D., McIntyre L. [32].	2020	Developing a framework to inform scale-up success for population health interventions: a critical interpretive synthesis of the literature	Global Health Research and Policy	Yes	3 phases, 11 actions and 4 key components for scaling up were identified by this framework.

**Table 2 ijerph-19-04773-t002:** Characteristics of the identified primary research studies. None were identified by gray literature search.

Authors	Year	Country	Study Setting	Intervention	Intervention Type	Based on Included Scaling-Up Frameworks
Croyden D.L., Vidgen H.A., Esdaile E., Hernandez E., Magarey A., Moores C.J., Daniels L. [33]	2018	Australia	Community-based setting, predominantly school venues	State-wide child obesity management program with the focus on families to help them lead healthier, happier lives by eating well and being more active.	Health Behavior Change	No
Del Díaz Castillo A., González S.A., Ríos A.P., Páez D.C., Torres A., Díaz M.P., et al. [34]	2017	Colombia	Public space (community setting, schools, health services, work sites)	Community-based programs offering active recreation and physical activity in public spaces.	Health Behavior Change	No
Herbert-Maul A., Abu-Omar K., Frahsa A., Streber A., Reimers A.K. [35]	2020	Germany	Community setting	A multifaceted community-based participatory research project aimed at promoting physical activity among women in difficult life situations.	Community-based Participatory Research	No
Hoelscher D.M., Kelder S.H., Murray N., Cribb P.W., Conroy J., Parcel G.S. [36]	2001	Texas	School setting	A multi-component, multiyear coordinated school health promotion program designed to decrease fat, saturated fat, and sodium in children’s diets, increase physical activity, and prevent tobacco use.	Health Behavior Change	No
Kennedy L., Pinkney S., Suleman S., Mâsse L.C., Naylor P.-J., Amed S. [37]	2019	Canada	Community setting	Sustainable Childhood Obesity Prevention through Community Engagement (SCOPE)	Multi-sectoral, Multi-component Community-based Participatory Intervention	No
Kozica S.L., Lombard C.B., Harrison C.L., Teede H.J. [38]	2016	Australia	Community setting	The Healthy Lifestyle Program is an evidence-based weight gain program for reproductive-age women, adapted for rural settings.	Health Behavior Change	No
McKay H.A., Macdonald H.M., Nettlefold L., Masse L.C., Day M., Naylor P.-J. [39]	2015	Canada	School setting	AS! BC! is a comprehensive school health-based model that provides teachers and schools with training and resources to integrate physical activity and healthy eating into the school environment.	Setting-related Intervention	No
Nigg C., Geller K., Adams P., Hamada M., Hwang P., Chung R. [40]	2012	Hawaii	(After) School setting	The Fun 5 program was integrated within the after-school setting with targets of at least three times per week of 30 min of physical activity and daily consumption of at least five fruit and vegetable servings (which was added in the first dissemination year).	Health Behavior Change	No
Sims-Gould J., McKay H.A., Hoy C.L., Nettlefold L., Gray S.M., Lau E.Y., Bauman A. [41]	2019	Canada	Community setting	CTM was a 6-month, choice-based, flexible, scalable, health promotion intervention for low active (< 150 min of moderate to vigorous PA/week) older adults (60+ years).	Health Behavior Change	No
Stewart A.L., Gillis D., Grossman M., Castrillo M., Pruitt L., McLellan B., Sperber N. [42]	2006	USA	Community setting	CHAMPS is a lifestyle PA program for older adults.	Health Behavior Change	No

## Data Availability

Not applicable.

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
