# Peer review of "Development of a Framework for Scaling Up Community-Based Health Promotion: A Best Fit Framework Synthesis"

_ijerph, 2022, doi:10.3390/ijerph19084773_

Round 1
Reviewer 1 Report
Thank you for the opportunity to review your paper which is a review of scaling up frameworks which you used to develop a new framework of components for informing a community and health promotion intervention project.
I think the actual approach to be quite acceptable, apart from some that and some questions on the method I make below and hope that some clarifications may improve the acceptability of the paper for publication.
A few things to note - I question that calling this a systematic review. I accept it being a systematic search and making it clear this is a ‘qualitative systematic review’, a narrative review or just a review. The Best fit method is for the synthesis of qualitative evidence but to call it a systematic review as per the title then it needs to be justified as a systematic review and refer to PRISMA guidelines.
The abstract is well written and sentences clear to follow but the structure of the remainder of the paper is difficult to follow, particularly the introduction, and need some reworking. The intro starts by discussing PA but then move to talk about health inequalities. The next paragraph speaks about community interventions but the focus seems to be directed towards participatory approaches – and I don’t see how this fits with the aim. Finally, you start to mention scaling up and the context of the KOMBINE project. The Introduction needs an extensive rewrite to focus on the key issues and reduce the confusing of ideas and information – I suggest you start by thinking about why each paragraph is needed and stick to one point per paragraph – if you break down what you mean in the summary as a succinct introduction then that would be a good approach.
The intro alludes to ‘wicked problems” a few times L53, L79 and L93 and 124 but doesn’t say what these “wicked problems” are. Please explain the context or reason. Issues like obesity or climate change are often quoted as wicked public health issues but I cant understand you connection between wicked problems (these or something else) and reviewing frameworks for scale up.
The intro discussion about the KOMBINE and VERBUND are confusing – I gather the purpose is to provide some context for why the review was important but maybe if that was clearer it would reduce the confusion as to why much of the introduction is focused on explaining it. Given you go back to it in the Discussion section I believe that is a better place for this discourse.
As an aside, why is health and physical activity promotion considered separately? Is this a community health program or a physical activity health promotion program or what is it?
You mention no other framework or model has focused on PA promotion with a focus on structural change – my understanding is that is what the PRACTIS model is all about. Please justify.
What are health inequalities interventions – I think the focus is on interventions that address health inequalities (which are/are not (??) necessarily PA interventions)? So does that mean the aims and method focusing on scaling up frameworks and interventions that have gone to scale are about health inequalities - if so this is unclear and the rationale does not make sense.
Method – The criteria for inclusion of frameworks does not appear sound or maybe just not well explained. If the focus is just on PA or just on health inequalities and there were multiple frameworks then it might make sense to look at just frameworks that had been used in that area however these are so few and it seems your KOMBINE project is about community health promotion not just PA. As such I am not sure I understand properly why you have excluded certain frameworks as you have – especially as the components developed are not exclusive for PA interventions and your discussion about these components seems to be broader.
Now having read your Discussion you make an important point L486-487 about the distinguishing features of your framework – sounds like that is a better reason for inclusion/exclusion of existing frameworks than whether they are about PA or health inequalities…
Results - Some components require some clarity. E.g. L309 what do you mean by ‘different roles occurring’ do mean different tasks require different roles or that different policy actors are involved in the process of scaling up (roles don’t ‘occur’). Personal skills – L320 – do you mean technical or personal skills here? Plan, conduct and apply assessment – do you mean research and evaluation?
Why is the political commitment and advocacy “evolving”? L368 is evolving commitment specifically something that was identified – you haven’t mentioned but if it is I think that is quite important to understand especially in the context of sustainability.
What do you mean to obtain collaboration? Build collaboration yes but obtain is an odd phrasing here too - L383.
L401 does encourage participation and ownership refer specifically to community partners or did you have something else in mind – most the research is on community partners but there is some others that maybe refer to research or government or industry partnering. Can you be explicit what is meant here form the research findings?
I like the discourse in L509-516 – this was important to highlight, though more could be said as to why current frameworks are insufficient for the purpose of understanding structural or system implementation upfront; and in the Discussion as to why this particular framework was needed/what benefits it provides over existing frameworks. Note this seems to be a different aim to that described in L443 of just providing an overview of existing frameworks.
Reviewer 2 Report
Thank you for the opportunity to review this interesting and necessary piece of work - it is timely, since we are now at a point where we do need to consider implementation of PA interventions, particularly in the context of scaling up those which already exist. There has been much reinvention of the wheel, so it is great to see we are now working toward a different, more sustainable approach.
The paper is well written and beyond a further proof to tighten up where possible, I have no specific comments about how the work is presented.
In the intro - the words 'wicked problems' are used. Would it be possible to use different terminology? As it currently reads, it does not sit well with me and I feel it is a little coercive.
Table 1 - would it be possible to add a column with detail to briefly outline the framework.
3.2 - please add more detail on how the themes were generated.
Fig 3 - lack indication of when each of these components should be implemented. You state that no chronological order was identified - but clearly there are things that happen before others. If you are not able to define the 'when', then please explicit how you came to this conclusion.
Characteristics table - please indicate which (if any) came from the gray literature search.
